# Evaluation of hip arthroscopy using a hip-specific distractor for the treatment of femoroacetabular impingement

**Tatiana Charles**[ID][☯], **Marc Jayankura**[ID]*[☯]

Department of Orthopaedic Surgery and Traumatology, Clinic of Hip and Pelvis, Hôpital Universitaire Erasme, Brussels, Belgium

☯ These authors contributed equally to this work.

* mjayanku@ulb.ac.be

**Data Availability Statement:** All relevant data are within the manuscript and its Supporting Information files.

**Funding:** The authors received no specific funding for this work.

## Abstract

### Background and study aims

Hip arthroscopy using an orthopaedic traction table has been associated with traction-related neurovascular complications. Since the use of a hip-specific distractor for performing hip arthroscopy hasn't been associated with those specific complications we hypothesized that a hip-specific distractor might facilitate the learning curve of hip arthroscopy for beginner surgeons.

### Material and methods

We reviewed retrospectively the first 56 hip arthroscopies performed to treat femoro-acetabular impingement using a hip-specific distractor. We tried to analyse the learning curve of this procedure using operative time, peri- and postoperative complications, hospital stay and patient satisfaction. We also evaluated pre- and postoperative sports activities and tried to identify some factors as poor postoperative prognostic factors.

### Results

Only 1 major complication occurred. No traction-related complications have been encountered. The curves analysing intervention time and postoperative satisfaction rate showed improvement after 30 cases performed. In all cases, we were able to perform the whole planned gesture without difficulties accessing the hip joint.

### Conclusion

The hip-specific distractor is a safe and reproducible method in performing hip arthroscopy without any traction-related complications or time limits.

**Competing interests:** The authors have declared that no competing interests exist.

## Introduction

Femoroacetabular impingement (FAI), defined as an abnormal contact between head and neck junction and the acetabular margin, is now widely recognized as a cause of major chondrolabral lesions leading to early osteoarthritis of the hip [1, 2]. Hip arthroscopy has become a gold standard in order to treat FAI in early stages [2, 3]. Yet, hip arthroscopy remains a difficult procedure due to a challenging learning curve, difficulties of accessing the hip joint and the confined work space [3–5]. Access to the hip joint could only be made possible by adequate hip distraction [1, 6].

Complication rates of hip arthroscopy vary between 0 and 13% [7]. Most of these complications are neurological injuries seen with distraction or compression on a traction table [7–10]. Distraction- and compression-type complications are related to the length of the procedure, thus limiting certain cases of hip arthroscopy in a timely manner [8]. These complications tend to decrease with the increasing experience of the surgeon [4, 9]. Another possible problem encountered when using a traction table in hip arthroscopy is the accessibility of the joint, as up to 18% of hip joints are described as being difficult accessible and even 2% being inaccessible [4, 8].

Traction forces up to 500 N are required to allow sufficient hip distraction [11]. Using an orthopaedic traction table, traction is applied along the entire lower limb thus losing much of the traction applied by distracting the knee joint (for about 5mm) and the ankle joint (for about 5mm) [11]. A hip-specific distractor applies about 500N of traction force exclusively on the hip joint [11]. The Schanz screws used are specially designed for this hip-specific distractor and the force applied to the hip joint can be calculated based on the elasticity of those screws. When about 500N of traction force is obtained, the Schanz screws will start to bend as seen on the fluoroscopic controls [11]. Additionally, acetabular anteversion induces a tendency of the femoral head to translate anteriorly, thus reducing the anterior joint space. This can be addressed by internal rotation of the femoral head putting the sciatic nerve at risk when using a posterolateral portal [11]. But a hip-specific distractor presents with a controlled antero-posterior traction vector in neutral rotation [11]. For those reasons the hip-specific distractor hasn't been associated with traction-related complications nor with difficulties accessing the hip joint, and could help beginner surgeons in performing hip arthroscopy [1, 11]. Therefore we hypothesized that the use of a hip-specific distractor might reduce complication rates such as neurovascular traction-related complications. The use of a hip-specific distractor might also decrease iatrogenic chondral and labral lesions when entering the joint because of better controlled joint opening, allowing surgeons to realize the entire planned procedure without time limits thus decreasing the learning curve in hip arthroscopy. The goal of this study was to evaluate the learning curve of hip arthroscopy using a hip-specific distractor and the complications related to this procedure.

## Material and methods

We performed a retrospective analysis of all successive patients who underwent a hip arthroscopy using a hip-specific distractor in order to treat FAI from September 2007 to March 2015. Hip arthroscopy was performed by a single surgeon (MJ), thus allowing the evaluation of this surgeon's learning curve. Hip arthroscopy with the use a hip-specific distractor is standard care at our institution.

All patients presented with a symptomatic FAI and a positive impingement test at the time of preoperative consultation. Preoperative work-up consisted of a standard X-ray of the pelvis and hip, arthro-CT and MRI or arthro-MRI of the hip. Classification of osteoarthritis was made using the Tönnis classification [12]. If the patients were sportsmen a total of 6 months of

conservative therapy was applied and in all other cases a total of 1 year of conservative treatment was completed. All patients who presented with a clinically and radiologically proven FAI and failed conservative treatment were considered for an arthroscopic procedure.

## Operative technique

The patient is positioned in lateral decubitus, opposite to the operated side (Fig 1), with a fluoroscopy placed at the level of the hip in order to control the placement of the hip-specific distractor and the arthroscopic portals. First, the hip-specific distractor (DR Medical AG, Solothurn, Switzerland) is placed. Two Schanz screws of 6mm diameter are positioned above the superior portion of the acetabulum (Fig 2) under fluoroscopic guidance. Two other screws are placed at the level of the lesser trochanter in the proximal femur. Screw placement technique is easy starting with blunt dissection and positioning of the tissue protecting sleeve to avoid any injury to surrounding soft tissues. Drilling is performed through the protecting sleeve followed by the insertion of the Schanz screw. The hip-specific distractor is then assembled allowing for an initial hip distraction followed by hip abduction (Figs 2 and 3). The Schanz screws (diameter of 6mm) were used to scale fluoroscopic images in order to be able to measure initial femoro-acetabular joint opening.

After proper hip distraction, a first anterolateral portal is positioned under fluoroscopic guidance using cannulated needles. The arthroscope is inserted through this first portal, allowing for a rapid examination of the hip joint. A second portal is then placed, 2 fingers breadth anterior to the first anterolateral portal under fluoroscopic and arthroscopic guidance. If

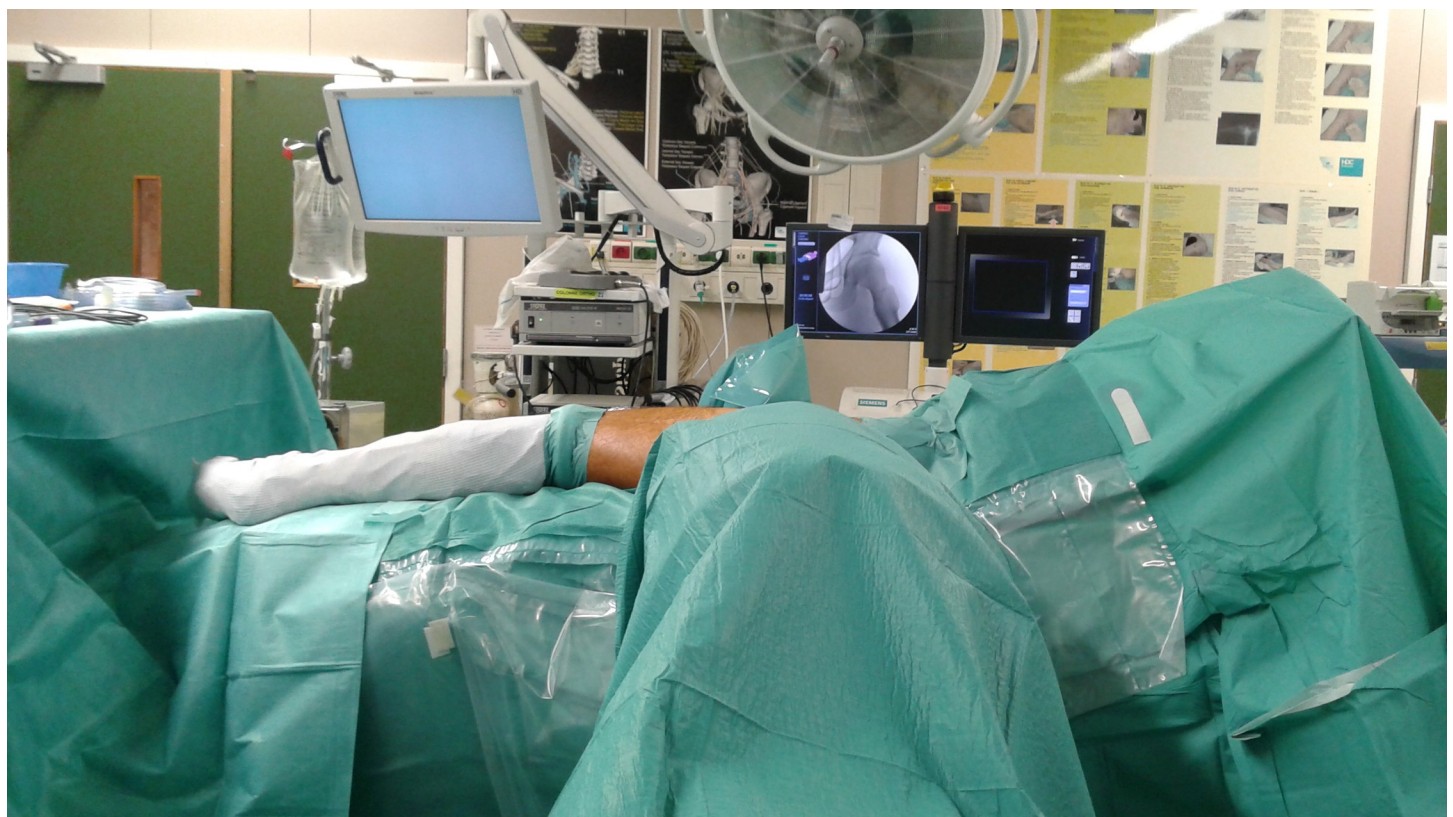

**Fig 1. Patient's positioning.** This picture shows the positioning of the patient on the operating table. The patient is positioned in lateral decubitus with the fluoroscopy in an arch around the hip of the patient.

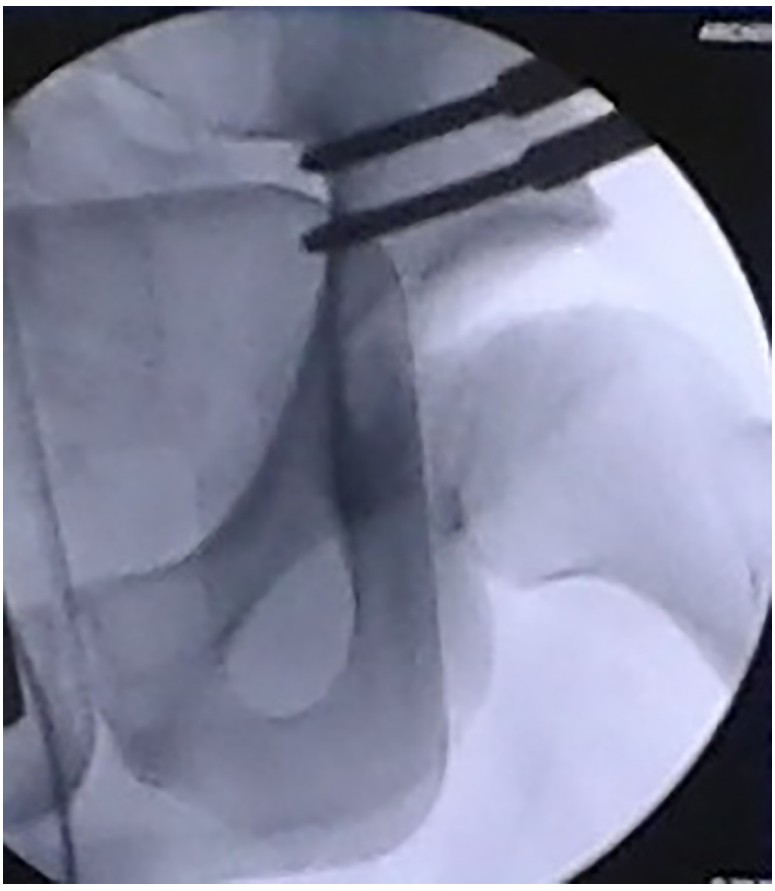

**Fig 2. Fluoroscopic control.** Fluoroscopic control (left hip, front view) after positioning of the hip-specific distractor representing the amount of hip distraction obtained.

necessary, the third portal can be placed posterior to the greater trochanter. After insertion of all 2 to 3 portals, we perform a short partial anterior curved capsulotomy between the first and the second portal. First the central compartment is inspected. In cases of associated chondral lesions debridement or microperforations are performed. Chondral lesions were classified according to the Outerbridge classification [13]. Labral lesions are assessed and debridement was performed if required. Then, osteoplasty of the acetabular rim is performed in cases of Pincer-type lesions. Finally reinsertion of instable labral lesions is performed.

After addressing the central compartment, distraction is progressively removed allowing for a thorough inspection of the peripheral compartment, and osteoplasty of the femoral head-neck junction is performed in cases of CAM-type lesions. After completely removing of the hip-specific distractor, the femoral neck osteoplasty is completed and we control arthroscopically and fluoroscopically for any residual impingement. The removal of the hip-specific distractor allows us to perform dynamic testing of the hip under arthroscopic visualisation to control the absence of any residual impingement. If an impingement remains present, further regularisation of the femoral neck can be performed.

The portals entries were infiltrated with a diluted ropivacaïne 2% for postoperative pain control before wound closure.

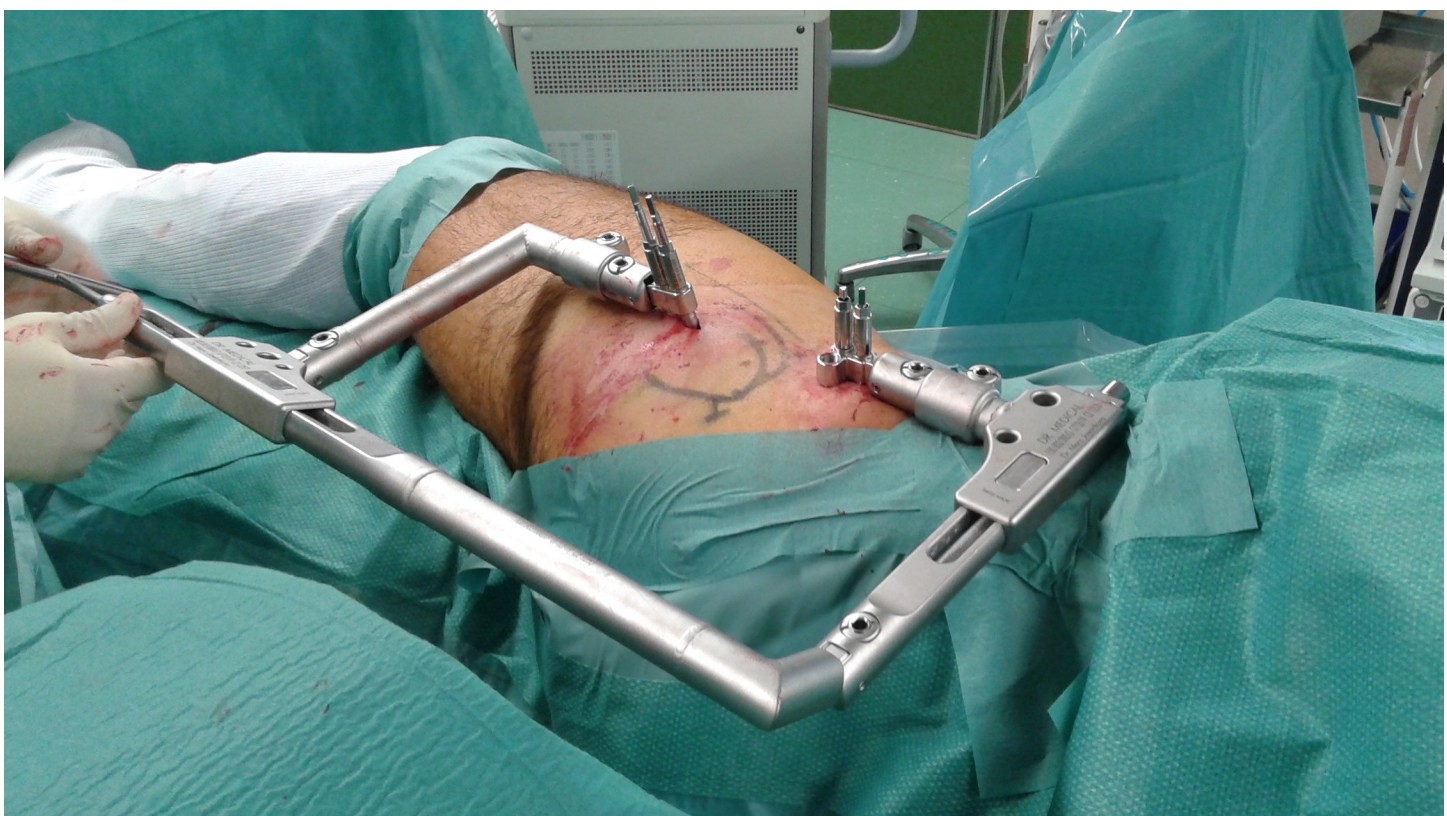

**Fig 3. Hip specific distractor.** This picture demonstrates the hip-specific distractor in place while distraction is applied.

During the postoperative period, patients were allowed a weight bearing up to 30kg for a period of 4 weeks associated with physiotherapy and indoor cycling to avoid capsular adhesions.

## Data analysis

The study was performed in 2 steps. During the first phase we analysed for objective values as the surgical intervention time, hospital stay, peri- and postoperative complication rates and conversion rate to total hip arthroplasty (THA). Patients were also evaluated for global satisfaction and their possibilities to return to sports if they were sportsmen. Analysis of any association regarding patient's satisfaction and THA conversion rate was performed with a Fisher's exact test, with a p-value < 0.05 considered to be statistically significant.

The second phase of the study was to try to analyse the learning curve of this procedure. We divided all cases by sequential groups of 10 patients each (Group A, B, C, D and E); the last group of patients containing a total of 16 patients as a group of only 6 patients was assumed to be too small to be able to be compared to the other groups. Then we were able to make curves based upon the intervention time, complication numbers, hospital stay, patient's satisfaction and conversion rate to THA. Differences in intervention time between the first 30 cases and the last 26 cases were also analysed. Data are given as mean ± SD. Homogeneity of variances was tested with a modified Levene test, and normality of the residuals was tested with the Shapiro-Wilks test. As these assumptions were rejected, the data were compared with the Mann-Whitney test. A global two-sided p-value <0.05 was considered statistically significant.

All statistical analyses were performed with the SAS University Edition (SAS Institute Inc., Cary, NC, USA) program. This study has been approved by the local committee of medical ethics at our institution (Comité d'éthique Erasme-ULB: P2017/483). Oral informed consent was obtained for all patients included in this study.

## Results

### General characteristics

Fifty-six hip arthroscopies were performed in 54 patients. There were 25 females and 31 males. Mean age of patients was 35 years old [17 – 59]. Six patients were lost of follow-up. Mean duration of follow-up was 53 months with a minimum of 12 months [12 – 101].

All patients benefitted preoperatively of a radiological work-up comprising conventional x-rays, MRI and arthro-CT showing signs of FAI in all symptomatic hips. In 26 hips isolated CAM-type lesions were identified whereas only 5 hips presented with isolated Pincer-type lesions. All other hips presented with signs of combined FAI. Thirty-three hips demonstrated no signs of osteoarthritis. Sixteen hips presented with a Tönnis grade 1 osteoarthritis, 6 hips with Tönnis grade 2 and 1 hip with a Tönnis grade 3 (S1 Table). Seven hips presented with at least 1 radiographic sign of acetabular retroversion. No patients presented with Developmental Dysplasia of the hip or sequelae of Legg-Calvé-Perthès disease.

### Surgery related data

Mean femoro-acetabular widening was 12mm [6 to 20]. Four of the five hips presenting with less than 10mm of hip distraction presented no signs of hip osteoarthritis on preoperative x-rays. Of those patients, only one patient was female, all other patients presenting with less than 10mm of hip distraction were males. In all cases the hip was found to be sufficient distracted to be able to perform hip arthroscopy. We encountered no difficulties accessing the hip joint. Overall mean surgical time was 223 minutes [120–485]. In all cases the entire surgical procedure could be performed as planned without any time limits. Mean hospital stay was 2 days postoperatively.

After intra-operative evaluation of the hips, 26 isolated osteoplasties of the femoral head-neck junction were performed, 5 isolated osteoplasties of the acetabular rim and 25 combined procedures were performed (Table 1). Labral lesions were debrided in 47 hips and reinsertion

**Table 1. Operative gesture.**

| Treatment of FAI | n [a] |
|---|---|
| Isolated Pincer lesions: Osteoplasty of acetabular rim | 5 |
| Isolated Cam lesions: Osteoplasty of femoral head-neck junction | 26 |
| Combined FAI: Combined procedure | 25 |
| **Treatment of associated labral lesions** | |
| Absence of lesions | 3 |
| Debridement | 47 |
| Reinsertion | 6 |
| **Treatment of associated chondral lesions** | |
| No supplementary gesture | 25 |
| Debridement | 22 |
| Microperforations | 9 |

Table 1 describes all operative gestures performed regarding the treatment of FAI and associated lesions as labral and chondral lesions.

[a] n = numbers of hips.

**Table 2. Complications.**

| Type of complications | n |
|---|---|
| **Major complications** | |
| Femoral fracture | 0 |
| Major bleeding | 0 |
| Complete dislocation | 0 |
| Hip subluxation | 1 |
| **Minor complications** | |
| Painful oedema of the torso (spontaneous resolution) | 2 |
| Minor postoperative wound bleeding (spontaneous resolution) | 1 |
| Minor iatrogenic head injury | 1 |
| Minor labral injury | 1 |

Table 2 summarizes all complications encountered in our series. Major complications were defined as complications which necessitated a prolonged hospital stay or reintervention. Minor complications were defined as complications without negative repercussions for the patient.

[a] n = number of hips.

could be performed in 6 hips. Interesting is that those labral reinsertion were only performed as from the 35th case in the learning curve. Acetabular chondral lesions were treated with debridement in 22 hips and with microperforations in 9 hips.

## Complication and total hip arthroplasty conversion rates

Total complication rate was 10,7% (Table 2). We encountered only 1 major complication represented by a hip subluxation. This patient benefitted of a THA 2 months following hip arthroscopy due to progressive hip dislocation and rapid osteoarthritis development (Fig 4). All other complications were minor complications and are summarized in Table 2.

The total conversion rate of THA was 17% (9 hips). Six patients benefitted of a THA during the first 2 years following hip arthroscopy for the treatment of FAI. One patient had a THA at 3 years, one at 4 years and a last patient at 6 years after arthroscopic treatment (S1 Table). Mean age of patient who had a conversion to THA was 39.4 years [24–53 y]. We found no association between the preoperative Tönnis classification and the conversion rate to THA (p-value 0.098)

## Satisfaction and return to sports

Global satisfaction rate of the patients in this series elevated up to 78%. When focussing on the group of younger patients ≤ 35 years (31 hips with a mean age of 27 years [17–33 y])

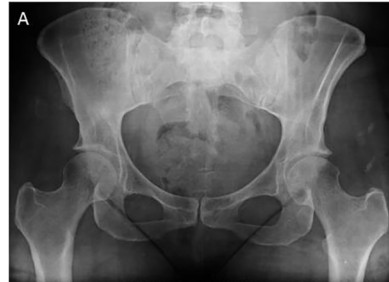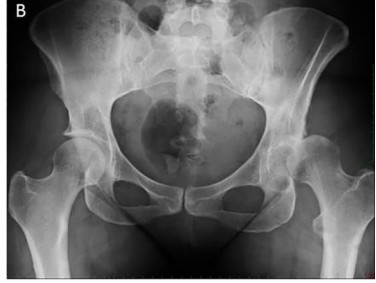

**Fig 4. Postoperative hip subluxation.** This picture shows (A) the preoperative x-rays and (B) the postoperative x-ray 1 month after hip arthroscopy with progressive hip dislocation and rapid osteoarthritis.

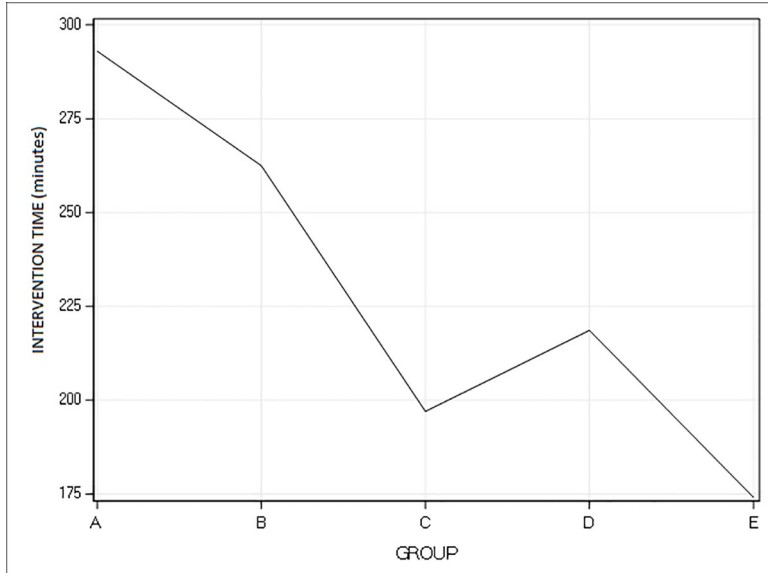

**Fig 5. Learning curve: Intervention time.** Representation of the decrease of intervention time in minutes (Y-axis) over the cases performed over time. The X-axis represents the five groups of patients with group A containing the first ten cases, group B the cases 11 to 20, group C the cases 21 to 30, group D the cases 31 to 40 and group E the cases 41 to 56.

satisfaction rate raised up to 89%. All patients who were satisfied assured us that they would undergo the same surgical procedure if they had to reconsider it. Even one patient who hadn't been satisfied mentioned he would do it all over again if he had the choice. We found no association between satisfaction rate and the Tönnis classification (p-value 0.337).

Twenty-six patients were sportsmen. Of those patients, 75% were able to return to their sports after the postoperative rehabilitation. And almost half of those patients (43%) were able to perform their sports at the same level as they performed before the start of their symptoms related to FAI. We found no association between satisfaction rate and whether or not patients were sportsmen (p-value 0.097).

## Learning curve

The curve based upon the intervention time (Fig 5) showed a good decrease in mean operating time over the cases performed. Mean intervention time for the first 30 cases was 250.8 ± 74.0 minutes compared to 191.1 ± 47.8 for the last 26 cases. This difference was statistically significant (p-value = 0.0002). The curve analysing patient's satisfaction showed also an improved satisfaction of patients of the cases performed. Over time complication rate and hospital stay remained stable.

## Discussion

In our series we noticed an overall decrease in operative time over time. A decrease of 22% of the total operative time has been noted between the first 30 cases and the last 26 cases of our series and this difference was found to be statistically significant (p < 0.05). This might be consistent with the cut-off described in the literature [4, 14].

Complication rate in our series remained quite stable over time. Although we had a change in nature of complications, as perioperative complications arose after the first 30 cases. This might reflect the increasing complexity of the interventions performed. Even if a decrease in

complication rate has been expected over time, other authors also have described a change in complications instead [7, 9]. This might be in accordance with the belief that while a surgeon gains experience and thus comfort in performing a particular intervention, he starts to operate more complex cases over time [7, 11, 14]. This might also be noted in our learning curve as mean intervention time of group D (patients 31 to 40) increased slightly compared to group C. This could be explained as labral repair, a more difficult and time-consuming procedure was started from the 35th case in the curve.

The overall complication rate (10.7%) in our series is similar as other complication rates described in the literature [1, 3, 7, 9]. We encountered only one major complication: a postoperative subluxation of the femoral head leading to an early conversion to THA. This complication has also turned up in another series and was thought to be related to an excessive resection of the anterior acetabular rim [7]. We believe this was also the case for this patient in our series who presented with postoperative subluxation of the hip, as an excessive acetabular resection induced a secondary hip instability with progressive dislocation and rapid hip osteoarthritis. We encountered only 1 minor postoperative complication probably related to the positioning of the patient on the operating table; a painful oedema of the torso which showed a spontaneous and early resolution in 2 patients.

Perioperative complications as labral and cartilage damage are thought to be underreported in the literature [1, 8, 10]. We encountered only 1 labral injury and 1 chondral injury in the late cases of our series. The labral injury might be due to a very stiff joint, as with sole distraction we were only able to achieve the minimal required distraction of the hip of 6mm to be able to start the procedure before capsulotomy. In the series of *Park et al.* those complications arose in the early and later phase of the learning curve [9]. As those lesions were supposed to be due to poor surgical technique and learning in the early phase, they were thought to be caused by stiff joints in the later phase [3]. Which might represent the gain in comfort of a surgeon in performing more difficult surgeries [7, 11, 14].

Most authors account the traction-related neurovascular complications as the most encountered complications following hip arthroscopy [7–10]. Traction-related complications can either be associated with the traction itself (distraction-type injury) or with the perineal post used as counter traction on orthopaedic traction tables (compression-type injury) [8]. Distraction-type injuries are described as transient neurapraxia, the sciatic and femoral nerves being the most vulnerable to prolonged and excessive traction [8]. Compression-type injuries are encountered in the area of the groin and can vary from pudendal nerve injuries to oedema, hematoma and pressure necrosis of the scrotum and labia majora [8]. Those complications can rise up to 10–15% [11] and are directly related to the technique, the length and the forces of the traction applied [5, 14, 15]. *Souza et al* found in their series that traction-related complications accounted for 58% of their total complication rate [7]. Another series presented a traction-related complication rate of 4% [4]. *Lee et al* described the presence of 3 traction-related complications in a series of 40 patients [5]. Traction-related complications are also described as being more frequently encountered during the first cases of a surgeon's learning curve and tend to decrease with the experience of this surgeon [4, 9]. In our series we encountered no traction-related neurovascular complications. Other series, describing the use of a hip-specific distractor, showed also the absence of neurovascular complications [1, 16]. On a traction table distraction is applied at the level of the foot and to obtain a 10 mm distraction at the hip to be able to perform hip arthroscopy, the additional lengthening of the ankle and the knee might lead to an overall lengthening of that lower limb potentially at risk for the sciatic nerve [1]. With the use of an hip-specific distractor, the traction applied is focused on the hip only, avoiding the need of traction on the whole limb and the use of a perineal post, thus avoiding

those traction-related complications both the distraction- and the compression-type injuries [8].

Up to 18% of hips are described as difficult accessible [8]. *Konan and al.* found 2% of hips inaccessible in their series [4]. In this series, representing a single surgeon learning curve, there were also 3 hips in which the surgical gesture couldn't be finished due to lack of time on the traction table [4]. Another author presented difficulties in starting the hip arthroscopy as 1.9% of the hips were found to be insufficient distracted to be able to start with the intra-articular compartment [3]. In those cases hip arthroscopy might be started through the peripheral instead of the central compartment [10]. In our series, all hips were found to be sufficient distracted to be able to perform hip arthroscopy, as planned starting with the intra-articular compartment. Also, in our experience, hip distraction was always increased after capsulotomy thus allowing for more working space. We could also perform the entire operative gesture in all cases as the use of a hip-specific distractor decreased the limit described when using the traction table [11]. This similarity has also been described using a hip-specific distractor in other series [1, 16].

Other potential positive aspects of a hip-specific hip distractor can be represented by the direct and immediate control that a surgeon has over the traction forces applied by the device [11]. External distraction can also easily be removed for dynamic testing of eventual residual impingement [11] which might be more difficult in our opinion on a traction table. Potential drawbacks for the use of a hip-specific distractor are the slightly increased operative time, as it takes approximately 18 minutes to position a hip-specific distractor [1]. In our personal experience, the time of positioning of the device can be decreased with the experience of the surgeon and good assistance. It is also more invasive than standard hip arthroscopy on a traction table [1, 8, 11]. Another question that can be raised is the potential conflict of the presence of the device while performing arthroscopic gestures. We never encountered any difficulties in performing the planned gestures while the hip-specific distractor was positioned. In our experience we also never had to abort an intervention nor change the planned intervention for lack of time. We personally believe this hip-specific distractor to be a very useful tool and therefore we continue to use this device when performing standard hip arthroscopy at our institution.

Although we were not able to decrease the learning curve of a surgeon using a hip-specific distractor, it might be helpful in the early phase as beginner surgeons wouldn't have to worry about distraction time and traction-related complications [11]. Another way to decrease early complications has been shown to be supervision [3]. *Dietrich et al* evaluated the learning curve of 2 different surgeons; the second surgeon being supervised by the first surgeon after his learning curve had been completed [3]. They showed a significant decrease in complication rates in the early cases performed by the second surgeon [3]. Therefore, the use of a hip-specific distractor might be useful for those beginner surgeons who could not possibly benefit of a guided supervision and help.

THA conversion rate in our series elevated up to 17%, which seems to be higher than those described in other series. *Philippon et al.* described a THA conversion rate of 9% after hip arthroscopy to treat FAI [17]. *Byrd et al.* described also a specific series of hip arthroscopy for Cam-lesions and showed a conversion rate to THA of 0.5% [15]. We believe that the THA conversion rate found in our series was related to a broader patient selection as some patients were 50 years and older of age. When analysing the curve we noticed that in the first 30 cases operated 7 out of 9 THA were performed, whereas only 2 patients were converted to THA in the last 26 cases performed. This could reflect the learning of the surgeon but it could also be biased as follow-up of the first cases performed was much longer than last cases.

The satisfaction rate of 78% in our series is comparable to results found in the literature. A small series described a treatment failure rate of 20% [5]. *Byrd et al.* showed also a general

postoperative improvement of 83% [15]. Another series using an invasive distractor showed an overall satisfaction rate of 79% [16].

In our series we were also able to notice a change in indication as the surgeon operated younger patients with less signs of osteoarthritis. This might also account for the increased satisfaction rate and decreased THA conversion rate in the later phase of our study.

We were not able to correlate a poor postoperative evolution with the presence of osteoarthritis. This might be due to the small number of patients in our series. Although, presence of osteoarthritis has been found to be a poor postoperative prognostic factor [16–19].

The strength of this study probably lies in the fact that all cases were performed by the same surgeon for the same operative indication at the same institution.

One of the major limitations of this study relies in its retrospective nature as we had a few losses of follow-up (10%) and lacked patient's satisfactions and evolutions. A second limitation of our study is represented by its relatively small size as only a total number of 56 cases were operated. This made the analysis of early evolution in a surgeon's experience possible but we were not able to evaluate his further progression as the curves didn't reach a real plateau yet.

## Conclusion

Hip arthroscopy for FAI using a hip-specific distractor is a safe and reproducible intervention with complications rate comparable to those found in the literature.

One of the major advantages of a hip-specific distractor is the absence of neurovascular traction-related complications with this technique. As those complications occur especially during the first part of a surgeon's learning curve [4, 9], a hip-specific distractor might be helpful at the beginning of a surgeon's experience as there is no time pressure and the procedure can be entirely performed as initially planned.

## Supporting information

**S1 Table.**
(DOCX)

## Author Contributions

**Investigation:** Tatiana Charles.

**Software:** Tatiana Charles.

**Supervision:** Marc Jayankura.

**Writing – original draft:** Tatiana Charles.

**Writing – review & editing:** Tatiana Charles, Marc Jayankura.

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
