## [Decision Letter · Decision Letter 0]

17 Aug 2020

PONE-D-20-12351

Advantages of hip arthroscopy using an external hip distractor for the treatment of femoroacetabular impingement.

PLOS ONE

Dear Dr. Jayankura,

Thank you for submitting your manuscript to PLOS ONE. After careful consideration, we feel that it has merit but does not fully meet PLOS ONE’s publication criteria as it currently stands. Therefore, we invite you to submit a revised version of the manuscript that addresses the points raised during the review process.

We look forward to receiving your revised manuscript.

Kind regards,

Osama Farouk

Academic Editor

PLOS ONE

Journal Requirements:

2. Please carefully proofread your manuscript for typographical errors. For example, in your methods section “We performed a retrospective analysis of all succesive patients …” should be “We performed a retrospective analysis of all successive patients.

Reviewers' comments:

Reviewer's Responses to Questions

**Comments to the Author**

1. Is the manuscript technically sound, and do the data support the conclusions?

Reviewer #1: Partly

Reviewer #2: Partly

2. Has the statistical analysis been performed appropriately and rigorously? 

Reviewer #1: I Don't Know

Reviewer #2: N/A

3. Have the authors made all data underlying the findings in their manuscript fully available?

Reviewer #1: No

Reviewer #2: Yes

4. Is the manuscript presented in an intelligible fashion and written in standard English?

Reviewer #1: Yes

Reviewer #2: Yes

5. Review Comments to the Author

Reviewer #1: Study Hypothesis: the use of Hip distractor for Hip arthroscopy was not associated with traction or compression complications, it would facilitate the learning curve. The aim of the study was to evaluate the learning curve of hip arthroscopy using a hip-specific distractor and the complications related to this procedure

The authors provided a good introduction stressing the fact that the use of external hip distractor during hip arthroscopy has not been associated with any of the traction or compression type complications, that otherwise could be associated with the use of traction table, hence They hypothesized that the use of a distractor would relieve the surgeon from the burden of Time stress during traction And will allow him completing the whole planned procedure without fear of traction or compression type complications.

They conducted a retrospective analysis with inclusion of all cases treated from September 2007 to March 2015 In a single center by a single surgeon. The operative technique is adequately described. They mention clear inclusion criteria.

Discussion of specific areas for improvement

Major points:

1. although the primary objective of the study was clearly defined, The descriptive and statistical analysis of the learning curve was inadequate.

the construction of such learning curve, how and why it was grouped into those unequal 5 groups is not clear in the text and was only mentioned in the figure 4 caption

2. Another major point is the lack of description of any statistical method or test used to evaluate changes in the learning curve

Other points to be considered

1. In the methodology, there is no mention of exclusion criteria. Were there any?

2. In the section of outcome measurement, no functional scores were collected perioperatively to objectively measure the outcome and only patient satisfaction (subjective) and total hip conversion rate were collected.( questionable value with short follow up which in some cases as short as one year)

3. The authors describe That all cases head preoperative imaging studies (X-rays, CT and MRI) however there is no reporting of the radiological parameters commonly used to measure The cam and pincer deformities whether pre- or postoperatively to control the adequacy of the performed surgery. This is important to support the claim of completeness of the surgical gesture in all the cases.

4. In the statistics section of the methodology, there is only mention of the used statistics software.

There is no mention of any statistical method or statistical test performed . Also whether the mentioned differences in the learning curve for intervention time and patient satisfaction were statistically significant or not.

5. also In the methodology, why selection was limited to the 56 hips although the authors described that until currently the learning curve is still ever changing. Was there power calculation for this sample size determined?

6. Line 111: labral reinsertion is performed after osteoplasty not vice versa.

7. The number of labral reinsertions in relation to the number of pincer/ mixed impingement treated is relatively low, here one would like to know where these more advanced time consuming reinsertions are located in the learning curve.

8. line 191: patients younger than 35, how many are they? A more detailed age description for example with the median value should be included.

9. line 192, 193 reconsider this sentence.

10. Concerning the 17% total hip arthroplasty conversion (9 hips) which was explained by the older age, here again the age of those patients should be mentioned (range and mean)

Secondly The cartilage condition intra operatively (grade and extent of cartilage damage) was not described this might be directly related.

Where in the learning curve are those cases located? persistence of symptoms, dissatisfaction and eventually total hip replacement could also occur in the beginning of learning curve, the adequacy of resections as mentioned before should be proven by radiological measurements?

Reviewer #2: Please review attached Doc in which I wrote all my requested changes

Thank you

Title - Advantages of hip arthroscopy using an external hip distractor for the treatment of femoro-acetabular impingement.

Comment 1 – looks irrelevant to the hypothesis and goal

“The goal of this study was to evaluate the learning curve of hip arthroscopy using a hip-specific distractor and the complications related to this procedure.” line 70

Removal/replacement of the word advantage with evaluation will reflect the idea better.

The word “advantage” draws the attention towards more benefits than the conventional traction table, however this is not the case here. Its only an evaluation of this traction technique.

Comment 2 – I prefer sticking to one name for the distractor used; either “external hip distractor” used in title or “hip-specific distractor” used in the introduction

Abstract

Comment 1 – word “per” should be replaced with peri

Comment 2 – poorly written. Needs to be re-written in a more attractive and informative way. Keeping in mind that this is the most read part of the paper. If not attractive, the reader will refrain from reading the paper.

Introduction

Generally, well-written covering the gap available in working with conventional traction table, and goal behind the study.

Comment 1 - Line 55: Distraction- of compression-type complications are related to the length of the……. Replace “of” with “and”

Comment 2 – I prefer sticking to one name for the distractor used; either “external hip distractor” used in title or “hip-specific distractor” used in the introduction

Materials and Methods

Generally, the materials and methods section lack the most important part the readers will be waiting for which is a precise and complete description of this specific distractor. Nothing was mentioned about the distractor apart form an intraoperative photo (Figure 3).

This is a major defect in this section that should be properly addressed under a specific subtitle together with proper illustrative photos or diagrams explaining its application and biomechanics.

Points of strength

• Type of study = retrospective September 2007 to March 2015

• Single disease = FAI

• Single Surgeon

Comment 1 – no reference was there for the preoperative conservative treatment protocol. On which basis this, 6m for athletes and 1-year conservative treatment protocols were set?

Comment 2 – Special mechanical analysis and attention should be paid and mentioned for the biomechanics of this distractor

? Material

? Line of forces pull, and Maximum force withstand

? Method of distraction in this instrument? Is it manual?

? Others, to be provided by manufacturer

Comment 3 – Lines 113,114 “Then, the peripheral compartment is assessed, and osteoplasty 113 of the femoral head-neck 114 junction is performed in cases of CAM-type lesions.”

I understood that you start working on the PC while traction is still applied… what is the advantage for this?

Comment 4 - Nothing was mentioned about the exclusion criteria or contraindication of using this distractor. Do you still use it in all patients? Even in osteopenic or osteoporotic bone?

Comment 5 – Supra- acetabular area is the pathway of the superior gluteal bundle supplying the gluteus medius, minimus and tensor facia lata muscles…. Did you have any injury for these important structures? I can see none in the results section…. Please describe your insertion technique to avoid these structures.

Results

Poor numbers for proper evaluation

• 56 hips were performed in 54 patients in 8 years

• 23 hips with OA

• 6 patients lost at follow up

Comment 1 – Line 154: Mean femoro-acetabular widening was 12mm [6 to 20].

The method of measurement was not described and verified in the M and M section. Please mention?

Comment 2 – Line 155: In all cases the hip was found to be sufficient distracted to be able to perform hip arthroscopy

Please verify and relate the distraction distances in different tonnis grades and different Genders?

Theses are the two main factors that affect the distraction power, these needs special results analysis and correlation.

Comment 3 - Complication and total hip arthroplasty conversion rates

This section is a critical section as its will be of main concern to the readers to understand the complications expected from this distractor.

1. I do not understand the major complication of the “hip subluxation”

Did it happen immediately postoperative or later? What was the expected reason? and why rapid conversion to THA? Were other treatment methods consumed?

I can see in the discussion section a brief analysis of this complication, however, no exact reason was detected or explained.

If available, please provide the postoperative radiology for this case, to share this abnormal complication with the readers.

2. There is no need to repeat the complications twice in paragraph and table format.

3. There is no need to mention the zero complications in the table format

Discussion

General unaddressed issue: as mention traction results into distraction and compression type problems. I can understand that this distractor avoids the compression problems, however, you mentioned that it avoids both despite using it without time limits!!!

Line 256- In our series we 256 encountered no traction-related neurovascular complications.

Line 63 could help beginner surgeons in performing 63 hip arthroscopy without concerning about the surgical time and thus traction-related 64 complications [1, 11].

I agree that traction here is focused on the hip as mentioned in the discussion but is this enough to avoid all the NV complication for distraction for anytime limit. I think time limit is still a very important factor and should never be underestimated. Please verify and discuss further….

The discussion is lacking the biomechanical values of the distractor. For example, the lines of pull it provides. We know that you need a lateral and distal line of pulls for the best hip asses and this is provided by putting the perineal post slightly laterally (not in midline) and adducting the hip to provide lateralization force with leg traction.

Can you explain and discuss similar pulling forces by the distractor and advantage/ disadvantage of it?

The discussion is focused on the surgeon learning curve, which I can see is not related to the device used. Please revise your discussion and focus it on the device used, which is the main issue here, explaining and discussing all aspects of its use.

6. PLOS authors have the option to publish the peer review history of their article (what does this mean?). If published, this will include your full peer review and any attached files.

Reviewer #1: **Yes: **Mohammad A. Masoud MD

Reviewer #2: No

---

## [Author Response · Author response to Decision Letter 0]

14 Oct 2020

Dear Editor and Members and Reviewers of the Editorial Board, 

Dear Editor in chief and reviewers, we thank you for the attention you’ve given our manuscript so far and we hope the changes we made to it will be to your liking. We’ve carefully read all your comments and we do hope that following answers and changes will help to clarify your decision. 

Journal Requirements: When submitting your revision, we need you to address these additional requirements.

a. Response: We controlled and ensured that our manuscript meets PLOS ONE’s style requirements, including those for file naming.

2. Please carefully proofread your manuscript for typographical errors. For example, in your methods section “We performed a retrospective analysis of all succesive patients …” should be “We performed a retrospective analysis of all successive patients.

a. Response: Our manuscript has been carefully proofread as asked for potential typographical errors.

a. Response: The corresponding author (MJ) has validated his ORCID iD in the Editorial Management as requested

a. Response: Caption for supporting file has been added. 

Reviewer #1: Study Hypothesis: the use of Hip distractor for Hip arthroscopy was not associated with traction or compression complications, it would facilitate the learning curve. The aim of the study was to evaluate the learning curve of hip arthroscopy using a hip-specific distractor and the complications related to this procedure

The authors provided a good introduction stressing the fact that the use of external hip distractor during hip arthroscopy has not been associated with any of the traction or compression type complications, that otherwise could be associated with the use of traction table, hence They hypothesized that the use of a distractor would relieve the surgeon from the burden of Time stress during traction And will allow him completing the whole planned procedure without fear of traction or compression type complications.

They conducted a retrospective analysis with inclusion of all cases treated from September 2007 to March 2015 In a single center by a single surgeon. The operative technique is adequately described. They mention clear inclusion criteria.

Discussion of specific areas for improvement

Major points:

1. although the primary objective of the study was clearly defined, The descriptive and statistical analysis of the learning curve was inadequate. the construction of such learning curve, how and why it was grouped into those unequal 5 groups is not clear in the text and was only mentioned in the figure 4 caption

Response: We adjusted the descriptive of the analysis of the learning curve as requested in the section material and methods. 

We adjusted also the part in material & methods describing the creation of the learning curves, but we didn’t perform any tests to evaluate changes as Hoppe et al stated in their systematic review that “Conceptually, a learning curve should be identified as a graph in which a consecutive number of cases are presented on the horizontal axis and some measure of proficiency or learning is presented on the vertical axis” (sic) http://dx.doi.org/10.1016/j.arthro.2013.11.012; Also testing between groups of 10 patients each might not clinically nor statically have a lot of meaning in our opinion. 

2. Another major point is the lack of description of any statistical method or test used to evaluate changes in the learning curve

Response: Also testing between groups of 10 patients each might not clinically nor statically have a lot of meaning in our opinion. (see response to previous point)

Other points to be considered

1. In the methodology, there is no mention of exclusion criteria. Were there any?

Response: No, there were no exclusion criteria, except for a patient’s possible refusal to participate to this study. A learning curve is the analysis of all successive patients who underwent a specific surgery for a specific indication, if there were other exclusion criteria, it might not reflect a proper learning curve. For example, if we had to exclude patients of older age, we might have had to exclude some of the first patients which would not reflect the actual learning of a surgeon in this specific setting.

2. In the section of outcome measurement, no functional scores were collected perioperatively to objectively measure the outcome and only patient satisfaction (subjective) and total hip conversion rate were collected.( questionable value with short follow up which in some cases as short as one year)

Response: Most learning curves limit themselves in analysing operative time and perioperative and postoperative complications. We actually also evaluated patient’s satisfaction rates and conversion/revision rates, which are parameters rarely evaluated when evaluating learning curves. The strong point of this study is that it actually evaluates the learning of hip arthroscopy for one specific indication, as some authors suggest that every gesture in hip arthroscopy might present with its personal learning curve. Indications are not mixed in this study. 

3. The authors describe That all cases head preoperative imaging studies (X-rays, CT and MRI) however there is no reporting of the radiological parameters commonly used to measure The cam and pincer deformities whether pre- or postoperatively to control the adequacy of the performed surgery. This is important to support the claim of completeness of the surgical gesture in all the cases.

Response: In the section results (subtitle general characteristics) preoperative radiological evaluation of all patients has been described in general. The claim of completeness is not in accordance of the whole CAM or the Whole pincer that has been corrected; I don’t think that anyone can claim that as easy. We claim that we could do all we planned to do, without concerning about time and traction-related complications. We were able to correct deformities, treat labral lesion and chondral lesion and control the absence of residual impingement with dynamic testing under fluoroscopic and arthroscopic visualisation. 

4. In the statistics section of the methodology, there is only mention of the used statistics software.

There is no mention of any statistical method or statistical test performed . Also whether the mentioned differences in the learning curve for intervention time and patient satisfaction were statistically significant or not.

Response: We adjusted the description of the statistics used in this manuscript.

5. also In the methodology, why selection was limited to the 56 hips although the authors described that until currently the learning curve is still ever changing. Was there power calculation for this sample size determined?

Response: No there was no power calculated for this study. When starting the study, we had only those first 56 patients who presented with at least 1 year of follow-up. We could not include others as follow-up was less than 12 months. 

6. Line 111: labral reinsertion is performed after osteoplasty not vice versa.

Response: Line 111: you are correct. We changed this; sorry for this mistake in the manuscript, we must have overlooked it

7. The number of labral reinsertions in relation to the number of pincer/ mixed impingement treated is relatively low, here one would like to know where these more advanced time consuming reinsertions are located in the learning curve.

Response: Where are those more time-consuming labral lesions situated in the learning curve? A very interesting question indeed. This has been added in the result section as those labral repairs were performed as from the 35th case and not earlier, thus slightly increasing mean intervention in the learning curve. I also took the liberty to add a note in the discussion regarding this.

8. line 191: patients younger than 35, how many are they? A more detailed age description for example with the median value should be included.

Response: Line 191: has been addressed in the revised manuscript.

9. line 192, 193 reconsider this sentence.

Response: Line 192: has been rewritten.

10. Concerning the 17% total hip arthroplasty conversion (9 hips) which was explained by the older age, here again the age of those patients should be mentioned (range and mean)

Secondly The cartilage condition intra operatively (grade and extent of cartilage damage) was not described this might be directly related.

Where in the learning curve are those cases located? persistence of symptoms, dissatisfaction and eventually total hip replacement could also occur in the beginning of learning curve, the adequacy of resections as mentioned before should be proven by radiological measurements?

Response: This has been addressed in the revised manuscript. 

Reviewer #2: Please review attached Doc in which I wrote all my requested changes

Thank you

Title - Advantages of hip arthroscopy using an external hip distractor for the treatment of femoro-acetabular impingement.

Comment 1 – looks irrelevant to the hypothesis and goal

“The goal of this study was to evaluate the learning curve of hip arthroscopy using a hip-specific distractor and the complications related to this procedure.” line 70

Removal/replacement of the word advantage with evaluation will reflect the idea better.

The word “advantage” draws the attention towards more benefits than the conventional traction table, however this is not the case here. Its only an evaluation of this traction technique.

Response: Title has been changed according as requested.

Comment 2 – I prefer sticking to one name for the distractor used; either “external hip distractor” used in title or “hip-specific distractor” used in the introduction

Response: As requested, we also paid attention and from now on, the only term referring to the device used is “hip-specific distractor”.

Abstract

Comment 1 – word “per” should be replaced with peri

Response: This has been changed. 

Comment 2 – poorly written. Needs to be re-written in a more attractive and informative way. Keeping in mind that this is the most read part of the paper. If not attractive, the reader will refrain from reading the paper.

Response: I tried to make the abstract more attractive and hope you will prefer that way. 

Introduction

Generally, well-written covering the gap available in working with conventional traction table, and goal behind the study.

Comment 1 - Line 55: Distraction- of compression-type complications are related to the length of the……. Replace “of” with “and”

Response: Has been changed.

Comment 2 – I prefer sticking to one name for the distractor used; either “external hip distractor” used in title or “hip-specific distractor” used in the introduction

Response: We changed every “external hip distractor” with the term “hip-specific distractor”, so we stuck to only 1 term referring to the device as requested. 

Materials and Methods

Generally, the materials and methods section lack the most important part the readers will be waiting for which is a precise and complete description of this specific distractor. Nothing was mentioned about the distractor apart form an intraoperative photo (Figure 3).

This is a major defect in this section that should be properly addressed under a specific subtitle together with proper illustrative photos or diagrams explaining its application and biomechanics.

Response: Regarding your comment of major defect in our section of material and methods…. I adjusted the introduction of the article, explaining a bit more about the line of work and how the hip-specific distractor works in a biomechanical way, so our hypothesis seems more concrete and better understandable for those who does not apply regularly this device. On the other hand, the main goal of this manuscript was not to describe the biomechanics of the hip-specific distractor but to describe a series (with learning curve) using the device. The biomechanics with diagrams as requested have already been described in other works published by other authors. (Sadri H. Techniques in hip arthroscopy and joint preservation surgery; Elsevier Saunders, 2011: pp 113-120)

Points of strength

• Type of study = retrospective September 2007 to March 2015

• Single disease = FAI

• Single Surgeon

Comment 1 – no reference was there for the preoperative conservative treatment protocol. On which basis this, 6m for athletes and 1-year conservative treatment protocols were set?

Response: No, the duration of conservative treatment before planning hip arthroscopy is still debated in the literature and not very clear. We know that some patients respond to conservative treatment hence surgery is not always warranted. But athletes tend to wait less before surgery because they need to restart their sports activities and cannot wait 1 year for conservative treatment to be completed followed by at least 6 months of rehabilitation after arthroscopic surgery.

Comment 2 – Special mechanical analysis and attention should be paid and mentioned for the biomechanics of this distractor

? Material

? Line of forces pull, and Maximum force withstand

? Method of distraction in this instrument? Is it manual?

? Others, to be provided by manufacturer

Response: See response to the same point raised above (I adjusted the introduction of the article, explaining a bit more about the line of work and how the hip-specific distractor works in a biomechanical way, so our hypothesis seems more concrete and better understandable for those who does not apply regularly this device. On the other hand the main goal of this manuscript was not to describe the biomechanics of the hip-specific distractor but to describe a series (with learning curve) using the device. The biomechanics with diagrams as requested have already been described in other works published by other authors.)

Comment 3 – Lines 113,114 “Then, the peripheral compartment is assessed, and osteoplasty 113 of the femoral head-neck 114 junction is performed in cases of CAM-type lesions.”

Response: you are correct. We changed this; sorry for this mistake in the manuscript, we must have overlooked it.

I understood that you start working on the PC while traction is still applied… what is the advantage for this?

Response: Distraction is progressively diminished when we start to work on the PC compartment. This has been adjusted. 

Comment 4 - Nothing was mentioned about the exclusion criteria or contraindication of using this distractor. Do you still use it in all patients? Even in osteopenic or osteoporotic bone?

Response: To answer your question, we still indeed use it in every case of hip arthroscopy we perform. We feel that the joint opening offered by the hip-specific distractor allows us to treat much better all lesions encountered in the joint. Nowadays, we can associate labral reinsertion with cartilage repair techniques with great ease, which we feel might be much more difficult without the hip-specific distractor. Since hip arthroscopy is mainly indicated in a young sportive population, osteoporotic bone is rarely a major concern.

Comment 5 – Supra- acetabular area is the pathway of the superior gluteal bundle supplying the gluteus medius, minimus and tensor facia lata muscles…. Did you have any injury for these important structures? I can see none in the results section…. Please describe your insertion technique to avoid these structures.

Response: Avoiding soft tissue damage: Placement of the schanz screws after blunt dissection and positioning of the protecting sleeve in the correct position. Drilling is performed through the protecting sleeve followed by screw insertion. This has been adjusted and added in the material and method, subsection ‘surgical technique’.

Results

Poor numbers for proper evaluation

• 56 hips were performed in 54 patients in 8 years

• 23 hips with OA

• 6 patients lost at follow up

Comment 1 – Line 154: Mean femoro-acetabular widening was 12mm [6 to 20].

The method of measurement was not described and verified in the M and M section. Please mention?

Response: The Schanz screws (diameter of 6mm) were used to scale fluoroscopic images in order to be able to measure femoro-acetabular joint opening. This has also been added in the material and methods to clarify our measurement methods. 

Comment 2 – Line 155: In all cases the hip was found to be sufficient distracted to be able to perform hip arthroscopy

Please verify and relate the distraction distances in different tonnis grades and different Genders?

Theses are the two main factors that affect the distraction power, these needs special results analysis and correlation.

Response: we reviewed those cases as asked and added this to the manuscript. But it is very disappointed as 4 of the 5 hips presented no signs of osteoarthritis and 4 of the hips were males.

Comment 3 - Complication and total hip arthroplasty conversion rates

This section is a critical section as its will be of main concern to the readers to understand the complications expected from this distractor.

1. I do not understand the major complication of the “hip subluxation”

Did it happen immediately postoperative or later? What was the expected reason? and why rapid conversion to THA? Were other treatment methods consumed?

I can see in the discussion section a brief analysis of this complication, however, no exact reason was detected or explained.

If available, please provide the postoperative radiology for this case, to share this abnormal complication with the readers.

Response: this complication has been further clarified in the manuscript as asked. We did modify this section so complications were not repeated both in text and table as asked. And we removed the “zero” complications of the table as asked, although this was to highlight the absence of traction-related complications.

2. There is no need to repeat the complications twice in paragraph and table format.

Response: This has been addressed in the revised manuscript. 

3. There is no need to mention the zero complications in the table format

Response: This has been addressed in the revised manuscript. 

Discussion

General unaddressed issue: as mention traction results into distraction and compression type problems. I can understand that this distractor avoids the compression problems, however, you mentioned that it avoids both despite using it without time limits!!!

Line 256- In our series we 256 encountered no traction-related neurovascular complications.

Line 63 could help beginner surgeons in performing 63 hip arthroscopy without concerning about the surgical time and thus traction-related 64 complications [1, 11].

I agree that traction here is focused on the hip as mentioned in the discussion but is this enough to avoid all the NV complication for distraction for anytime limit. I think time limit is still a very important factor and should never be underestimated. Please verify and discuss further….

Response: Regarding you concern about time limit and potential NV complications. Up to now, all series describing the use of a hip-specific distractor have never been associated with traction-related complications regardless of the intervention time as seen in series using an orthopaedic traction table. I admit, not a lot of surgeons do use a hip-specific distractor as some might think it less attractive due to its more invasiveness than an orthopaedic traction table, so there are not a lot of studies describing the use of hip-specific distractor, which is one of the reasons why this one might be very interesting. But its use has never been associated with traction-related complications regardless of the intervention time. I don’t think that this needs further description in the discussion section. Nevertheless, I do agree that time is a very important factor. A hip arthroscopy should not take a whole day to be performed, for the patient’s sake. 

The discussion is lacking the biomechanical values of the distractor. For example, the lines of pull it provides. We know that you need a lateral and distal line of pulls for the best hip asses and this is provided by putting the perineal post slightly laterally (not in midline) and adducting the hip to provide lateralization force with leg traction.

Can you explain and discuss similar pulling forces by the distractor and advantage/ disadvantage of it?

Response: This is a retrospective case series describing the use of the distractor. It is not a biomechanical study evaluating the feasibility of the hip-specific distractor. Others authors have already described this. (Also see previous response regarding similar comment.) But I don’t think that a text-book description of the biomechanics of the hip-specific distractor has its place in the discussion of this study.

The discussion is focused on the surgeon learning curve, which I can see is not related to the device used. Please revise your discussion and focus it on the device used, which is the main issue here, explaining and discussing all aspects of its use.

Response: Personally, I don’t understand why we should revise the entire discussion, since this study is describing “a learning curve”, it might be normal that it is compared to other published learning curves in order to see if there might be potential advantages in using this device. Throughout the entire first part (1 page) of the discussion we discuss complications associated or not with the device, which is very important since avoidance of traction-related complication is one of the major reasons why a hip-specific distractor is so interesting. Lines 281 to 293 are dedicated to the pros and cons of the device. Repeating the description of the surgical technique has no necessity in the discussion in our opinion. 

Thank you for considering our manuscript, 

Thank you for your revision and comments, they were very helpful. 

Sincerely, 

The authors, 

Marc Jayankura & Tatiana Charles

---

## [Decision Letter · Decision Letter 1]

30 Nov 2020

PONE-D-20-12351R1

Evaluation of hip arthroscopy using a hip-specific distractor for the treatment of femoroacetabular impingement.

PLOS ONE

Dear Dr. Jayankura,

Thank you for submitting your manuscript to PLOS ONE. After careful consideration, we feel that it has merit but does not fully meet PLOS ONE’s publication criteria as it currently stands. Therefore, we invite you to submit a revised version of the manuscript that addresses the points raised during the review process.

We look forward to receiving your revised manuscript.

Kind regards,

Osama Farouk

Academic Editor

PLOS ONE

Reviewers' comments:

Reviewer's Responses to Questions

**Comments to the Author**

1. If the authors have adequately addressed your comments raised in a previous round of review and you feel that this manuscript is now acceptable for publication, you may indicate that here to bypass the “Comments to the Author” section, enter your conflict of interest statement in the “Confidential to Editor” section, and submit your "Accept" recommendation.

Reviewer #1: (No Response)

2. Is the manuscript technically sound, and do the data support the conclusions?

Reviewer #1: Partly

3. Has the statistical analysis been performed appropriately and rigorously? 

Reviewer #1: No

4. Have the authors made all data underlying the findings in their manuscript fully available?

Reviewer #1: Yes

5. Is the manuscript presented in an intelligible fashion and written in standard English?

Reviewer #1: Yes

6. Review Comments to the Author

Reviewer #1: 1- Response to point 1: the statement by Hoppe et al in their systematic review that “Conceptually, a learning curve should be identified as a graph in which a consecutive number of cases are presented on the horizontal axis and some measure of proficiency or learning is presented on the vertical axis” (sic)

http://dx.doi.org/10.1016/j.arthro.2013.11.012; does not contradict a statistical significance testing. In the same article, a cut off of 30 cases could be identified, this could have been used in this study to make 2 bigger groups for testing instead of testing between groups of 10 patients each with little meaning.

The authors in the current study describe already a 22% improvement in intervention time between the first 30 cases and the following cases. Is this difference statistically significant?

2- Again Concerning the 17% total hip arthroplasty conversion (9 hips) the authors in their revision gave more information about the age of this group, however, they did not mention how these cases plotted on the learning curve and again the cartilage condition intra operatively (grade and extent of cartilage damage) still needs to be described.

Some of the important causes of total hip conversion could be either a high Tönnis grade of arthritis preoperative, extensive cartilage damage or inadequate FAI-correction or even over-correction which is common in the beginning of learning curve.

where in the learning curve are those cases located was still not mentioned in the revised manuscript. Unfortunately, the authors could not provide a perioperative radiological evidence of adequate deformity correction. For example, a perioperative alpha angle correction and some of the acetabular measures as the CE angle should be provided.

3- Minor spelling errors:

• Line 30 correct to performed

• Line 53 Complications: remove S

• Line 96 proven FAI and failed conservative treatment where considered for an arthroscopic procedure: correct to were?

• Line 116 hip-specific external hip distractor: remove 2nd hip

• Lines 128, 129: Labral lesions are assessed and debridement if required. The sentence is not complete, correct to : Labral lesions are assessed and debridement was performed if required.

• Line 154 to analyse de learning curve: correct to the ?

7. PLOS authors have the option to publish the peer review history of their article (what does this mean?). If published, this will include your full peer review and any attached files.

Reviewer #1: **Yes: **Mohammad Dr. Masoud

---

## [Author Response · Author response to Decision Letter 1]

4 Dec 2020

Subject Letter:

 Response to reviewers

Dear Editor and Members and Reviewers of the Editorial Board, 

Dear Editor in chief and reviewers, we thank you for the attention you’ve given our manuscript so far and we hope the changes we made to it will be to your liking. We’ve carefully read all your comments and we do hope that following answers and changes will help to clarify your decision. 

Response to Reviewer #1: 

1- Response to point 1: the statement by Hoppe et al in their systematic review that “Conceptually, a learning curve should be identified as a graph in which a consecutive number of cases are presented on the horizontal axis and some measure of proficiency or learning is presented on the vertical axis” (sic) http://dx.doi.org/10.1016/j.arthro.2013.11.012; does not contradict a statistical significance testing. In the same article, a cut off of 30 cases could be identified; this could have been used in this study to make 2 bigger groups for testing instead of testing between groups of 10 patients each with little meaning.

The authors in the current study describe already a 22% improvement in intervention time between the first 30 cases and the following cases. Is this difference statistically significant?

Response: As requested we calculated whether or not this difference of intervention time between the first 30 cases and the last 26 cases was statistically significant. It was statistically significant with a p-value of 0.0002. Statistical tests applied and results were added in the material & methods section as well as in the result section. 

2- Again Concerning the 17% total hip arthroplasty conversion (9 hips) the authors in their revision gave more information about the age of this group, however, they did not mention how these cases plotted on the learning curve and again the cartilage condition intra operatively (grade and extent of cartilage damage) still needs to be described.

Some of the important causes of total hip conversion could be either a high Tönnis grade of arthritis preoperative, extensive cartilage damage or inadequate FAI-correction or even over-correction which is common in the beginning of learning curve.

where in the learning curve are those cases located was still not mentioned in the revised manuscript. Unfortunately, the authors could not provide a perioperative radiological evidence of adequate deformity correction. For example, a perioperative alpha angle correction and some of the acetabular measures as the CE angle should be provided.

Response: In the result section describing THA conversion rates we also refer to table S1 with complementary results representing all cases performed (from the first case to the last). The table shows exactly where those cases of THA conversions are situated in the learning curve as well as the exact operative gestures performed during hip arthroscopy. We added a column resuming acetabular chondropathy graded during the intervention. We also added a few comments on the subject in the hope it clarifies this further for the readers in the discussion section. 

3- Minor spelling errors

• Line 30 correct to performed

Response: This has been addressed in the revised version.

• Line 53 Complications: remove S

Response: This has been addressed in the revised version.

• Line 96 proven FAI and failed conservative treatment where considered for an arthroscopic procedure: correct to were?

Response: This has been addressed in the revised version.

• Line 116 hip-specific external hip distractor: remove 2nd hip

Response: This has been addressed in the revised version.

• Lines 128, 129: Labral lesions are assessed and debridement if required. The sentence is not complete, correct to : Labral lesions are assessed and debridement was performed if required.

Response: This has been addressed in the revised version.

• Line 154 to analyse de learning curve: correct to the ?

Response: This has been addressed in the revised version.

Thank you for considering our manuscript, 

Thank you for your revision and comments, they were very helpful. 

Sincerely, 

The authors, 

Marc Jayankura & Tatiana Charles

---

## [Decision Letter · Decision Letter 2]

11 Jan 2021

PONE-D-20-12351R2

Evaluation of hip arthroscopy using a hip-specific distractor for the treatment of femoroacetabular impingement.

PLOS ONE

Dear Dr. Jayankura,

Thank you for submitting your manuscript to PLOS ONE. After careful consideration, we feel that it has merit but does not fully meet PLOS ONE’s publication criteria as it currently stands. Therefore, we invite you to submit a revised version of the manuscript that addresses the points raised during the review process.

We look forward to receiving your revised manuscript.

Kind regards,

Osama Farouk

Academic Editor

PLOS ONE

Reviewers' comments:

Reviewer's Responses to Questions

**Comments to the Author**

1. If the authors have adequately addressed your comments raised in a previous round of review and you feel that this manuscript is now acceptable for publication, you may indicate that here to bypass the “Comments to the Author” section, enter your conflict of interest statement in the “Confidential to Editor” section, and submit your "Accept" recommendation.

Reviewer #1: All comments have been addressed

2. Is the manuscript technically sound, and do the data support the conclusions?

Reviewer #1: Yes

3. Has the statistical analysis been performed appropriately and rigorously? 

Reviewer #1: Yes

4. Have the authors made all data underlying the findings in their manuscript fully available?

Reviewer #1: Yes

5. Is the manuscript presented in an intelligible fashion and written in standard English?

Reviewer #1: Yes

6. Review Comments to the Author

Reviewer #1: Thank you for the conducted corrections.

My single comment here has to do with the supplementary materials table. I had to guess that the (redo?) Column means if the patient would accept to have the surgery again when asked in the follow up. Is this correct. Please clarify this so the reader doesn't mistake this for revision rate or something.

7. PLOS authors have the option to publish the peer review history of their article (what does this mean?). If published, this will include your full peer review and any attached files.

Reviewer #1: **Yes: **Mohammad Masoud

---

## [Author Response · Author response to Decision Letter 2]

14 Jan 2021

Dear Editor and Members and Reviewers of the Editorial Board, 

Dear Editor in chief and reviewers, we thank you for the attention you’ve given our manuscript so far and we hope the changes we made to it will be to your liking. We’ve carefully read all your comments and we do hope that following answers and changes will help to clarify your decision. 

Response to Reviewer #1: 

My single comment here has to do with the supplementary materials table. I had to guess that the (redo?) Column means if the patient would accept to have the surgery again when asked in the follow up. Is this correct. Please clarify this so the reader doesn't mistake this for revision rate or something.

Response: We changed the name of the column from “Redo?” to “Do it again?” and added a comment to clarify this column in the legend below the table. 

Thank you for considering our manuscript, 

Thank you for your revision and comments, they were very helpful. It was a real pleasure and honour to collaborate with you for our manuscript. 

Sincerely, 

The authors, 

Marc Jayankura & Tatiana Charles

---

## [Editor Report · Decision Letter 3]

18 Jan 2021

PONE-D-20-12351R3

Evaluation of hip arthroscopy using a hip-specific distractor for the treatment of femoroacetabular impingement.

PLOS ONE

Dear Dr. Jayankura,

Thank you for submitting your manuscript to PLOS ONE. After careful consideration, we feel that it has merit but does not fully meet PLOS ONE’s publication criteria as it currently stands. Therefore, we invite you to submit a revised version of the manuscript that addresses the points raised during the review process.

ACADEMIC EDITOR: Please address the reviewer's comment.

We look forward to receiving your revised manuscript.

Kind regards,

Osama Farouk

Academic Editor

PLOS ONE

Additional Editor Comments (if provided):

Please address the reviewer's comment "My single comment here has to do with the supplementary materials table. I had to guess that the (redo?) Column means if the patient would accept to have the surgery again when asked in the follow up. Is this correct. Please clarify this so the reader doesn't mistake this for revision rate or something."

---

## [Author Response · Author response to Decision Letter 3]

20 Jan 2021

Dear Editor and Members and Reviewers of the Editorial Board, 

Dear Editor in chief and reviewers, we thank you for the attention you’ve given our manuscript so far and we hope the changes we made to it will be to your liking. We’ve carefully read all your comments and we do hope that following answers and changes will help to clarify your decision. 

Response to Reviewer #1: 

My single comment here has to do with the supplementary materials table. I had to guess that the (redo?) Column means if the patient would accept to have the surgery again when asked in the follow up. Is this correct. Please clarify this so the reader doesn't mistake this for revision rate or something.

Response: Yes, you are correct indeed. We changed the name of the column from “Redo?” to “Do it again?” and added a comment to clarify this column in the legend below the table. We hope this will clarify this further for the readers. 

Thank you for considering our manuscript, 

Thank you for your revision and comments, they were very helpful. It was a real pleasure and honour to collaborate with you for our manuscript. 

Sincerely, 

The authors, 

Marc Jayankura & Tatiana Charles

---

## [Editor Report · Decision Letter 4]

25 Jan 2021

Evaluation of hip arthroscopy using a hip-specific distractor for the treatment of femoroacetabular impingement.

PONE-D-20-12351R4

Dear Dr. Jayankura,

We’re pleased to inform you that your manuscript has been judged scientifically suitable for publication and will be formally accepted for publication once it meets all outstanding technical requirements.

Kind regards,

Osama Farouk

Academic Editor

PLOS ONE
---

## [Editor Report · Acceptance letter]

1 Feb 2021

PONE-D-20-12351R4 

Evaluation of hip arthroscopy using a hip-specific distractor for the treatment of femoroacetabular impingement. 

Dear Dr. Jayankura:

I'm pleased to inform you that your manuscript has been deemed suitable for publication in PLOS ONE. Congratulations! Your manuscript is now with our production department. 

Kind regards, 

on behalf of

Dr. Osama Farouk 

Academic Editor

PLOS ONE